# Sustaining Healthy Habits: The Enduring Impact of Combined School–Family Interventions on Consuming Sugar-Sweetened Beverages among Pilot Chinese Schoolchildren

**DOI:** 10.3390/nu16070953

**Published:** 2024-03-26

**Authors:** Chenchen Wang, Yijia Chen, Hao Xu, Weiwei Wang, Hairong Zhou, Qiannan Sun, Xin Hong, Jinkou Zhao

**Affiliations:** 1Department of Non-Communicable Disease Prevention, Nanjing Municipal Center for Disease Control and Prevention, Nanjing 210003, China; isisccwang@163.com (C.W.); m15005655732@163.com (Y.C.); 18375314830@163.com (H.X.); nj_wangww@126.com (W.W.); zhouhrong@126.com (H.Z.); elizabeth1002@163.com (Q.S.); 2Department of Non-Communicable Disease Prevention, Jiangsu Provincial Center for Disease Control and Prevention, Nanjing 210009, China

**Keywords:** sugar-sweetened beverages, school-based intervention, family-based intervention, schoolchildren, behavior sustaining, China

## Abstract

This study assesses the enduring impact of combined school- and family-based interventions on reducing the consumption of sugar-sweetened beverages (SSBs) among schoolchildren in China. Two primary schools were assigned at random to either the Intervention Group or the Control Group, in Nanjing, eastern China. All students were in grade three and received an invitation to participate. In the first year, students in the Intervention Group received one-year intervention measures, including monthly monitoring, aiming to decrease the consumption of SSBs. Students in the Control Group only received regular monitoring without interventions. In the second year, both groups received only regular monitoring, without active interventions. A generalized estimating equations model (GEE) was used to assess the intervention effects. After two years, relative to the Control Group, the Intervention Group had a significantly improved knowledge of SSBs and an improved family environment with parents. In the Intervention Group, 477 students (97.3%) had adequate knowledge about SSBs, compared to 302 students (83.2%) in the Control Group (X^2^ = 52.708, *p* < 0.001). Two years later, the number of students who stated ‘my home always has SSBs’ in the Intervention Group (7.8%) was fewer than that in the Control Group (12.4%), which was a statistically significant finding (*p* < 0.05). One year later, both the frequency and the quantity of SSB consumption in the Intervention Group were less than those in the Control Group; such differences between the groups remained statistically significant for the quantity but not for the frequency of SSB consumption two years later. In the Intervention Group, the frequency of SSB consumption was significantly reduced by 1.0 times per week, compared to a reduction of 0.1 times per week in the Control Group in the first year (*p* < 0.05). In the second year, the frequency of SSB consumption was reduced by 0.8 times per week in the Intervention Group, compared to 0.5 times per week in the Control Group (*p* > 0.05). In the first year, the volume of SSB consumption was significantly reduced by 233 mL per week in the Intervention Group, compared to an increase of 107 mL per week in the Control Group (*p* < 0.05). In the second year, the volume of SSB consumption was reduced by 122 mL per week in the Intervention Group compared to an increase of 31 mL per week in the Control Group (*p* > 0.05). The combined school-based and family-based interventions had a positive effect on the students’ knowledge of SSBs and their family dynamics during the first and second year. Relative to the Control Group, the Intervention Group had a statistically significant reduction in SSB consumption after 1 year, but not after 2 years.

## 1. Introduction

The consumption of sugar-sweetened beverages (SSBs) among children has increased globally [1]. Across all age groups, children are among the largest consumers of SSBs [2,3,4,5,6,7,8]. In the United States, 80% of youth consume SSBs regularly, contributing to approximately 11% of their daily energy intake [3]. In Australia, more than 50% of the free sugar intake in adolescent diets comes from SSBs [4]. In Korea, 27.7% middle-school students consume carbonated beverages > 3 per week [5]. Similarly, in China, reports have pointed to a rapid increase in SSB production and SSB consumption among children in recent years [6,7,8]; an earlier study indicated that 61.9% students aged 6–17 years drink SSBs at least once every week [7].

SSBs encompass non-alcoholic beverages sweetened with sugar, including carbonated beverages, fruit/vegetable drinks, lactobacillus or milk-based beverages, sweetened tea, and similar variants [1,2]. SSBs are the leading source of added sugar in the diet. For instance, a typical 12 fl. oz (355 mL) serving of carbonated beverages delivers 35.0–37.5 g of added sugar with 140–150 calories [9]. A robust body of evidence has linked the habitual consumption of excess amounts of SSBs in children and adolescents with dental caries, weight gain, and a higher risk of chronic disease through weight gain in adulthood. These observations make SSBs a clear target for policy and regulatory actions [9,10,11].

Childhood and adolescence are key periods of rapid physical and mental growth, during which individuals often become more active physically, and therefore consume more snacks and beverages that contain sugar [12,13]. Childhood and adolescence are also critical periods for cultivating healthy dietary behaviors [13]. In 2006, the Nutrition Friendly Schools Initiative was introduced by the World Health Organization, emphasizing comprehensive measures for school-based nutrition and health education [14]. Over the past few years, there has been a growing availability of public health interventions to reduce SSB consumption in children [15,16,17,18]. Schools, with their health education mandate, connections to families and communities, and access to substantial numbers of children over extended periods, represent compelling settings for public health nutrition interventions targeting children and adolescents [16,17]. Studies from various countries provide evidence that health and nutrition education at schools can effectively enhance nutritional knowledge in children [16,17,18,19,20]. Teo et al. conducted a school nutrition intervention for three months in primary schoolchildren aged 7–11 years in Malaysia, indicating the integration of nutrition education with a health-oriented school canteen environment demonstrated positive effects on eating behaviors [19]. Children, in contrast to adults, lack flexibility to choose the environment in which they live and drink beverages. Previous studies have also shown that a collaborative effort between children and parents to decrease SSB consumption could be a strategy for families [17,20].

These school-based or family-based interventions adopted educational and behavioral approaches that centered on enhancing children’s knowledge, attitudes, and, subsequently, their behaviors towards SSB consumption [21]. According to the evaluation immediately after the interventions, these interventions appeared to be effective in the short term [21,22]. In a previous study by Ebbeling et al., the experimental group received a 1-year intervention designed to decrease their consumption of SSBs, with follow-up for an additional year without intervention; the study showed that increases in BMI were smaller in the experimental group than in the control group after a 1-year intervention, but not at the 2-year follow-up [16]. A cluster-controlled trial, implemented in two primary schools between 2017 and 2019 in China, showed that the students’ knowledge about SSB consumption decreases to some degree following the cessation of interventions [23]. Poor self-discipline and strong peer influence among children may explain the non-sustained effect of interventions. It may also be influenced by family and environmental factors [24,25,26]. In the literature, there is a lack of evidence concerning how the healthy behaviors formed during intensive interventions can be sustained for a longer period of time in the absence of those interventions.

We hypothesized that after the one year intervention, in the Intervention Group, the knowledge about SSBs would be sustained at a level higher than baseline values and SSB consumption would remain lower than baseline values, while acknowledging there may be a rebound after one year of intervention. Therefore, this study aimed to assess the long-term impact of the school-based and family-based interventions on reducing the SSB consumption of Chinese primary schoolchildren in pilot schools from an eastern region in China.

## 2. Subjects and Methods

### 2.1. Participants

This study included two primary schools situated in Lishui district, Nanjing, eastern China, from September 2019 to September 2021. These two schools were similar in size, school facilities, and the composition of students. Every third-grade student in two schools received an invitation to participate in the present study. This study’s objectives and procedures were thoroughly explained to all participants, and written informed consent was obtained from a parent or a caregiver. This study was approved by the Academic and Ethical Committee, Nanjing Municipal Center for Disease Control and Prevention (CDC) (No. 2019-002) and registered at the Chinese Clinical Trial Registry with a number of ChiCTR2000033945.

### 2.2. Study Design

Two schools were assigned either to the Intervention Group or the Control Group by a computerized random drawing method. In the first year, students in the Intervention Group provided consent to receive 1-year intervention measures to reduce the consumption of SSBs and accepted monthly monitoring. Those in the Control Group only received the regular monitoring as received by those in the Intervention Group. In the second year, both groups received only regular monitoring, without any active interventions. The Control Group was granted access to the program resources and support once follow-up data collection was complete at the end of the second year.

### 2.3. Intervention Methods

The intervention measures encompassed both school-based and family-based measures to reduce SSB consumption. The school-based interventions consisted of four components, (a) monthly health education courses delivered as a 15-min video, (b) support within the school environment involving a display of eight posters in public places, (c) class environment support, focusing on monthly updates to the class bulletin board by students in line with monthly classroom sessions, and (d) implementation of a sugar-free campaign, including free drinking water and prohibition of SSB sales on campus. Family-based interventions incorporated components from the following list: (a) Health lectures, offered to the parents to promote reduced SSB consumption once per semester; (b) Monthly delivery of core messages through social media, such as parent WeChat group and parents’ self-supported QQ group, including in the school breaks during summer and winter. The core messages were designed and delivered by the study team, in accordance with contents of the in-classroom monthly health education sessions; (c) The use of ‘little hands holding big hands’, the distribution of intervention booklets to schoolchildren, encouraging interactive activities between students and parents at home; and (d) Collaborative efforts between schoolchildren and parents to create themed tabloids or paintings focused on ‘Reducing SSBs’ each semester. The implementation of both family-based and school-based interventions was verified through photographic evidence and videos, ensuring the transparent documentation of activities such as health education sessions, poster displays, parent lectures, and social media interactions.

### 2.4. Data Collection

The identical questionnaire was self-administered by participating students at baseline in September 2019, at the end of the first year in September 2020, and at the end of the second year in September 2021. The administration was facilitated by a standard PowerPoint slide set in each of the classrooms, involving both the Intervention and Control Groups. The study team, comprised of a class teacher, a local CDC member of staff, and a health care teacher, underwent training conducted by the principal investigator. The questionnaire covered demographic details such as date of birth, sex, grade, class, along with inquiries about knowledge regarding SSBs, the family environment regarding SSBs, the weekly accumulated physical activity outside school, daily cumulative screen time, homework time, weekly frequency of SSB consumption, and average intake per occurrence.

The SSB knowledge assessment comprised 10 questions, with correct answers obtained from eight health education courses. Each correct response earned 1 point, while an incorrect or ‘I don’t know’ response received 0 points. Individuals scoring 6 or higher were categorized as possessing ‘adequate knowledge about SSBs.’

The questionnaire for the family environment included six questions covering aspects such as family beverage consumption, parental attitudes, and behaviors related to SSBs. Students indicated their stance on each statement by selecting either ‘disagree’ or ‘agree’.

The weekly SSB intake frequency questionnaire was administrated to obtain information on the schoolchildren’s consumption of beverages. Eight broad beverage categories were grouped based on nutrient content and on China’s Beverage General Rule (GB10789-2008 [27]), including carbonated drinks, fruit/vegetable drinks, sweetened tea drinks, coffee drinks, milk drinks, sports drinks, plant-protein drinks, and brewed drinks. The consumption of SSBs was measured on a 7-day frequency scale, using the question ‘How many times did you drink carbonated beverages in past week, commonly available in the market?’. The standard PowerPoint listed eight different types of common beverage packaging and pictures in the market to assist students in the identification. The frequency of SSB consumption was determined based on the reported drinking of any of the eight categories. The overall quantity of SSBs consumed was computed by multiplying the reported frequency by the average amount drunk each time.

Parental education level, along with body weight and height, was self-reported by parents when obtaining the informed consent. Trained nurses measured the height and weight of the children at baseline, at the end of the first and second years. The measuring instruments were regularly calibrated following standard protocols. Height and weight were recorded to the nearest 0.1 cm or kg, respectively. Body mass index (BMI) was then calculated using the formula BMI = weight (kg)/height (m)^2^.

### 2.5. Statistical Analysis

The analyses were conducted using IBM SPSS Statistics Version 23.0 (SPSS Inc., Chicago, IL, USA). Descriptive analysis for qualitative variables included absolute frequencies and percentages. Quantitative variables with a normal distribution are described as a mean with standard deviation (SD). Pearson’s chi-square test was applied to compare categorical variables. An independent *t* test was utilized for normally distributed continuous variables. Given the three consecutive years of measurements for each subject, recognizing the lack of independence among these measures, generalized estimating equations models (GEE) were employed to assess outcomes related to SSB knowledge, family environment concerning SSBs, and weekly SSB intake. This approach accounts for the correction between the repeated measures taken on the same participants. GEE models were adjusted for child age, sex, parents’ education level, father’s BMI, mother’s BMI, physical activity time outside school (min/week), homework time (min/day), and screen time (min/day) at baseline. A *p* value below 0.05 was considered statistically significant.

## 3. Results

### 3.1. Demographic and Clinical Characteristics

At baseline, a total of 911 participants, comprising 533 from the Intervention Group and 378 from the Control Group, were initially enrolled and successfully completed the assessment. Among the participants enrolled at baseline, 490 individuals in the Intervention Group and 363 in the Control Group completed the trial and outcome assessments at the end of both the first and second years, included as the cohort in the final analysis. The retention rate for study participants was 97.8% at 1 year and 93.6% at 2 years (Figure 1). The main reasons for dropping out include absenteeism at school on the day of the outcome assessment and students’ migration to another district for school. Between participating subjects and those who dropped out, there were no observed differences in socio-demographic characteristics (Table 1).

Table 2 presents the demographic characteristics of the participants. The mean age was 9.1 years (SD ± 0.4) for 470 (55.1%) boys and 383 (44.9%) girls. More than 60% of parents had university educational attainment; 56.4% fathers and 21.3% mothers were overweight or obese; 40.1% of participants had weekly physical activity time outside school ≥ 120 min/week; 47.8% of participants had daily homework time ≥ 120 min/day; 73.2% of participants had daily screen time < 60 min/day. No significant differences in demographic characteristics were observed between the participants from the Intervention Group and Control Group (Table 2).

### 3.2. Knowledge about SSBs

Prior to the intervention, 254 (51.8%) and 212 (58.4%) students in the Intervention Group and Control Group had adequate knowledge about SSBs, respectively (X^2^ = 3.626, *p* = 0.061). One year later, 464 students (94.7%) and 296 students (81.5%) had adequate knowledge about SSBs in the Intervention Group and the Control Group, respectively (X^2^ = 37.127, *p* < 0.001). Two years later, 477 students (97.3%) in the Intervention Group and 302 students (83.2%) in the Control Group had adequate knowledge about SSBs (X^2^ = 52.708, *p* < 0.001). Compared to the baseline, two groups had increasing trends in having adequate knowledge about SSBs over time (first year, Intervention Group X^2^ = 182.719, *p* < 0.001, Control Group X^2^ = 64.239, *p* < 0.001; second year, Intervention Group X^2^ = 146.612, *p* < 0.001, Control Group X^2^ = 71.304, *p* < 0.001), but the Intervention Group exhibited a greater percentage of students with adequate knowledge about SSBs compared to the Control Group. Following the intervention, the proportion of students in the Intervention Group answering each of the 10 SSB knowledge questions correctly was significantly higher than in the Control Group (all *p* < 0.05) (Table 3).

### 3.3. Differences in Family Environment with Parents Concerning SSBs

Table 4 presents the differences between the Intervention and Control Groups, from baseline to the second year in the family environment with parents concerning SSBs. At baseline, the harms of SSBs were flagged to the parents in the Intervention Group and they were more in the ‘yes’ responses than those in the Control Group. No significant differences were observed in various aspects of the family environment between the Intervention Group and Control Group (all *p* > 0.05). One year later, the responses indicating ‘my home always has SSBs’ in the Intervention Group (7.3%) were less than those in the Control Group (13.5%). Two years later, in addition to ‘my parents restricted me from drinking SSBs’, the family influence of parents and at home of SSBs was more positive in the Intervention Group than in the Control Group. The GEE models showed that in the Intervention Group, in addition to ‘my parents often eat sugary snacks’, the family influence of parents and at home of SSBs was more positive in the first year and second year with statistical significance (all *p* < 0.05) (Table 4).

### 3.4. Frequency and Quantity of SSB Consumption

Table 5 presents the differences between the Intervention and Control groups from baseline to the second year in frequency and quantity of SSBs consumed. At baseline, SSB consumption in the Intervention Group was not significantly different from that in the Control Group (all *p* > 0.05). One year later, both the frequency (2.4 ± 2.7 times/week versus 3.4 ± 3.0 times/week) and quantity (638 ± 822 mL/week versus 1015 ± 1065 mL/week) of SSB consumption in the Intervention Group were less than those in the Control Group. Two years later, the frequency (2.7 ± 3.1 times/week versus 3.0 ± 3.2 times/week) of SSB consumption in the Intervention Group was not significantly different from that in the Control Group (*p* > 0.05), and the quantity of SSB consumption in the Intervention Group was lower than that in the Control Group (748 ± 1125 mL/week versus 939 ± 1304 mL/week) (*p* < 0.05). In the second year, the frequency of SSB consumption reduced by 0.8 times per week in the Intervention Group and 0.5 times per week in the Control Group, with no statistical significance. The volume of SSB consumption significantly reduced by 233 mL per week in the Intervention Group, compared to an increase of 107 mL per week in the Control Group in the first year (*p* < 0.05), and then decreased by 122 mL per week versus an increase of 31 mL per week in the second year (*p* > 0.05). The GEE models showed that in the Intervention Group, the frequency of SSB consumption decreased in the first and second year, while the quantity of SSB consumption decreased in the first year but not in the second year (Table 5).

## 4. Discussion

To the best of our knowledge, this represents the inaugural examination of the sustained impact of integrated school-based and family-based interventions on reducing the consumption of SSBs among Chinese primary schoolchildren in pilot schools [16,17,18,19,28]. Our findings indicate that this comprehensive approach had a favorable influence on SSB knowledge and the family environment of SSBs, leading to a reduction in the frequency and quantity of SSB consumption during the initial year, only to rebound by the conclusion of the second year. Relative to the Control group, the Intervention Group had a statistically significant reduction in SSB consumption in terms of both frequency and quantity after 1 year, but not for the frequency after 2 years.

Knowledge is the basis of practice formation, and only when knowledge rises to the level of belief can an individual find it possible to adopt a positive attitude to the practice of change [22,23,29,30]. In our study, the combined school-based and family-based interventions had a positive effect on the knowledge about SSBs after the 1-year intervention and follow-up at 2 years among schoolchildren. In comparison to the baseline, both groups exhibited a rising trend in acquiring adequate knowledge about SSBs over time. Notably, the Intervention Group displayed a higher proportion of schoolchildren with adequate knowledge about SSBs compared to the Control Group. In recent years, most nutrition education interventions, carried out among students, have effectively improved schoolchildren’s knowledge [29,30]. Our results are consistent with the study by Xu et al., in which students in the intervention schools were given 2 years of health and nutrition education courses from 2018 to 2020 in three counties of China [13]. In our study, the SSB knowledge score was increased by 1.0 and 0.6 points, respectively, in the first and second years in the Intervention Group. This indicates that schools, where students spend eight or more hours daily on weekdays, continue to be among the most effective settings for students to gain nutritional knowledge concerning SSBs [31,32].

Our study found that the family influence of parents at home relating to SSBs were positive in the first year and second year. The family-based intervention measures also exerted a positive impact on parents’ behavior, including reducing the storage of SSBs at home, cautioning their children about the adverse effects of SSBs, imposing restrictions on their children’s SSB consumption, and fostering a reduction in parental SSB intake. Our findings are consistent with previous reviews that found multi-component school-based nutrition interventions targeting children tended to be more effective in reducing SSB consumption when they included a parent-targeted component [33,34]. Parents’ involvement may be particularly important in improving the impact of these interventions. They not only play a significant role in shaping children’s choices and consumption but also serve as primary role models for children’s beverage choices and drinking behavior, especially among primary school students [35,36]. Future interventions to promote students’ healthy behaviors may consider incorporating strategies to appropriately engage parents [37,38,39].

Our study reveals a noteworthy decline in the frequency and quantity of SSB consumption in the first year. Specifically, there was a significant reduction in frequency of once per week in the first year and the quantity of SSBs consumed saw a notable decrease of 233 mL per week in the first year. This suggests that our school-based and family-based combined interventions had a positive impact on modifying SSB consumption behaviors following a one-year intervention. These findings align with the research by Zhu et al., supporting the effectiveness of one-year school-based interventions in reducing SSB consumption among Chinese students aged 9–14 years [40]. Additionally, Ebbeling et al. demonstrated in their study that a one-year intervention reducing SSB consumption resulted in a virtual elimination of reported SSB intake among overweight and obese adolescents, with persistent effects on diet observed through the 2-year follow-up [16].

Notably, compared to the Control Group, our study found a reduction in the frequency and quantity of SSB consumption in the Intervention Group, statistically significant in the first year, but not in the second year. Despite the effectiveness of our intervention in enhancing students’ knowledge about SSBs, we observed a widening discordance between knowledge and practice over time. In our study, the school-based and family-based interventions adopted educational and behavioral approaches that focused on improving children’s knowledge, attitude, and, subsequently, their behaviors towards SSB consumption. No mandatory interventions were in place to prohibit the consumption of SSBs. This phenomenon echoes the findings of Sligo et al., who noted that students may be aware of health-improving approaches but may face obstacles in translating that awareness into practice [41,42]. A review of interventions targeting SSB intake suggests that a duration of at least 12 months is necessary to induce substantial changes in student behavior [43]. Therefore, future research should consider additional interventions to reduce the consumption of SSBs, based on improved knowledge, to further narrow the discordance between knowledge and practice and sustainably reduce SSB intake. This adjustment may address the challenge of maintaining positive behavioral changes over time [44,45].

Our results provide some evidence regarding school-based and family-based interventions in the setting of an eastern environment and culture. Our study also provided a sustainable environment for reducing SSB consumption. For instance, all the schools participating in our program were equipped with direct drinking water machines, which were cleaned and disinfected regularly. Free drinking water was available on campus to all students all the time. The project schools also issued a long-standing policy document prohibiting the sale of SSBs on campus. It is expected that the adoption and sustaining of healthy behaviors would be facilitated through a more conducive environment and enforced by relevant policy.

The present study has several limitations. First, the sample of this study was from one district in Nanjing, which may not represent all primary schoolchildren in China. Second, randomization was conducted at the primary school level rather than at the individual level. Third, the SSB consumption among the schoolchildren was self-reported, introducing the possibility of reporting, social desirability or recall biases, especially among 9–10-year-old children, with a potential for underreporting. Fourth, we did not collect students’ daily total food intake and body measurements other than height and weight. We could not therefore assess the diet quality or health outcomes. Fifth, the Control Group received regular monitoring without active interventions in the first year; however, they may have been influenced by the Intervention Group’s changes in behavior. The study team made every effort to reduce contamination; however, there may still be possibilities of contamination given the short distance between the two schools and very dynamic interpersonal interactions and communications. Despite these limitations, our study stands out as one of the few endeavors in China that has successfully combined school-based and family-based interventions to address students’ SSB consumption. It contributes valuable insights that should be taken into account when designing future interventions aimed at reducing SSB consumption among students.

## 5. Conclusions

In conclusion, our study assessing the long-term impact of combined school- and family-based interventions on reducing SSB consumption among primary schoolchildren in pilot schools in China reveals positive effects on the knowledge about SSBs, and family dynamics, persisting into the second year following a one-year intervention. Our study also provides crucial insights for the design of future interventions in China, particularly in the integration of family components to foster sustained positive dietary behaviors among schoolchildren. Future research should consider extending the duration or intensifying the intervention to further narrow the discordance between knowledge and practice and sustainably reduce SSB intake.

## Figures and Tables

**Figure 1 nutrients-16-00953-f001:**
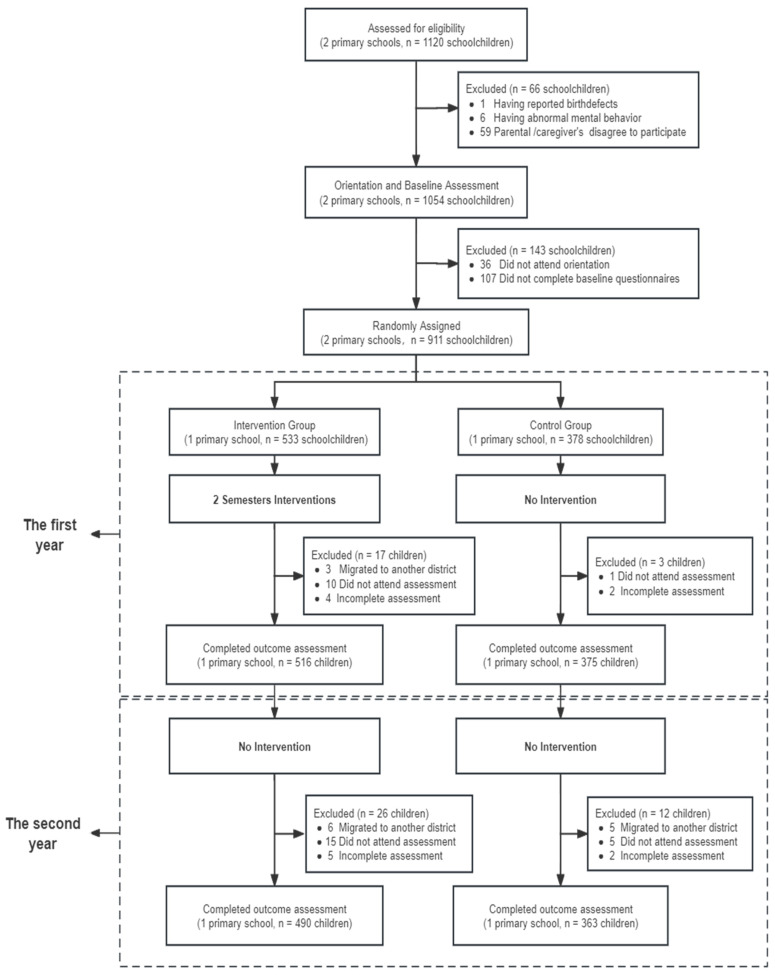
Screening, randomization, and follow-up of the study participants.

**Table 1 nutrients-16-00953-t001:** Demographic characteristics between schoolchildren retained in trial and dropped out (N/%, Mean ± SD).

Characteristics	Retained in Trial (853)	Dropped Out (58)	X^2^/t	*p*
Sex				
Male	467/54.7	32/55.2	0.004	0.950
Female	386/45.3	26/44.8		
Age (years)	9.1 ± 0.4	9.1 ± 0.3	0.685	0.494
Group				
Intervention Group	490/57.4	34/58.6	0.031	0.861
Control Group	363/42.6	24/41.4		
Parental education level				
≤9 years	336/39.4	22/37.9	0.048	0.826
>9 years	517/60.6	36/62.1		
Father’s BMI ^a^ (kg/m^2^)				
<24	370/43.6	25/43.1	0.005	0.943
≥24	479/56.4	33/56.9		
Mother’s BMI ^a^ (kg/m^2^)				
<24	664/78.7	45/77.6	0.038	0.845
≥24	180/21.3	13/22.4		
Physical activity time outside school (min/week)				
<120	511/59.9	30/51.7	1.507	0.220
≥120	342/40.1	28/48.3		
Homework time (min/day)				
<120	444/52.2	26/44.8	1.193	0.275
≥120	406/47.8	32/55.2		
Screen time (min/day)				
<60	624/73.2	48/82.8	2.589	0.108
≥60	229/26.8	10/17.2		
BMI ^a^	17.4 ± 2.9	17.6 ± 2.9	−0.722	0.470
Frequency of SSB ^b^ consumption	3.5 ± 4.0	3.5 ± 3.2	0.049	0.961

^a.^ BMI (body mass index) = weight (kg)/height (m)^2^. ^b.^ SSBs = sugar-sweetened beverages.

**Table 2 nutrients-16-00953-t002:** Demographic characteristics of participants (N/%, Mean ± SD).

Characteristics	Total (853)	Intervention Group (*n* = 490)	Control Group (*n* = 363)	X^2^/t	*p*
Sex					
Male	470/55.1	274/55.9	196/54.0	0.312	0.577
Female	383/44.9	216/44.1	167/46.0		
Age (years)	9.1 ± 0.4	9.1 ± 0.3	9.1 ± 0.4	−1.565	0.118
Parental education level					
≤9 years	336/39.4	169/34.5	167/46.0	11.583	0.001
>9 years	517/60.6	321/65.5	196/54.0		
Father’s BMI ^a^ (kg/m^2^)					
<24	370/43.6	208/42.5	162/45.0	0.512	0.474
≥24	479/56.4	281/57.5	198/55.0		
Mother’s BMI ^a^ (kg/m^2^)					
<24	664/78.7	375/77.0	289/81.0	1.916	0.166
≥24	180/21.3	112/23.0	68/19.0		
Physical activity time outside school (min/week)					
<120	511/59.9	305/62.2	206/56.7	2.622	0.120
≥120	342/40.1	185/37.8	157/43.3		
Homework time (min/day)					
<120	444/52.2	263/53.7	181/50.3	0.959	0.332
≥120	406/47.8	227/46.3	179/49.7		
Screen time (min/day)					
<60	624/73.2	352/71.8	272/74.9	1.017	0.348
≥60	229/26.8	138/28.2	91/25.1		
BMI ^a^	17.4 ± 2.9	17.3 ± 2.9	17.5 ± 3.0	−1.163	0.245
Frequency of SSB ^b^ consumption	3.5 ± 4.0	3.5 ± 4.4	3.5 ± 3.2	−0.144	0.885

^a.^ BMI (body mass index) = weight (kg)/height (m)^2^. ^b.^ SSBs = sugar-sweetened beverages.

**Table 3 nutrients-16-00953-t003:** Schoolchildren’s knowledge about SSBs between the Intervention Group and Control Group before and after intervention (N ^a^/%).

Knowledge	Baseline	First Year	Second Year	Change from Baseline		Compared to Baseline ^c^					
				First Year	Second Year	β_1_ ^d^ (95%CI)	Wald_χ2_	*p*	β_2_ ^e^ (95%CI)	Wald_χ2_	*p*
Definition of SSBs ^b^.											
Intervention Group	127/25.9	363/74.1	384/78.4	236/48.2	257/52.5	2.1 (1.8, 2.4)	211.855	<0.001	2.4 (2.1, 2.6)	288.113	<0.001
Control Group	129/35.5	175/48.2	196/54.0	46/12.7	67/18.5	0.5 (0.2, 0.8)	12.913	<0.001	0.8 (0.5, 1.0)	26.653	<0.001
*p*	0.003	<0.001	<0.001								
SSBs are bad for health.											
Intervention Group	432/88.2	485/99.0	480/98.0	53/10.8	48/9.8	2.6 (1.7, 3.5)	31.229	<0.001	1.9 (1.2, 2.5)	28.369	<0.001
Control Group	342/94.2	348/95.9	356/98.1	6/1.7	14/3.9	0.4 (−0.3, 1.1)	0.992	0.319	1.1 (0.3, 2.0)	6.414	0.011
*p*	0.003	0.005	0.908								
SSBs may cause tooth decay.											
Intervention Group	407/83.1	480/98.0	477/97.3	73/14.9	70/14.2	2.3 (1.6, 2.9)	48.728	<0.001	2.0 (1.4, 2.6)	46.499	<0.001
Control Group	321/88.4	337/92.8	336/92.6	16/4.4	15/4.2	0.5 (0.1, 1.0)	5.522	0.019	0.5 (0.0, 0.9)	4.756	0.029
*p*	0.031	<0.001	0.002								
SSBs may cause childhood overweight and obesity.											
Intervention Group	340/69.4	469/95.7	475/96.9	129/26.3	135/27.5	2.3 (1.8, 2.7)	95.181	<0.001	2.6 (2.1, 3.2)	94.124	<0.001
Control Group	266/73.3	320/88.2	319/87.9	54/14.9	53/14.6	1.0 (0.7, 1.3)	39.511	<0.001	1.0 (0.7, 1.3)	37.507	<0.001
*p*	0.223	<0.001	<0.001								
SSBs can increase type 2 diabetes in children and in later life.											
Intervention Group	208/42.4	419/85.5	427/87.1	211/43.1	219/44.7	2.1 (1.8, 2.4)	193.840	<0.001	2.2 (1.9, 2.5)	176.556	<0.001
Control Group	181/49.9	216/59.5	207/57.0	35/9.6	26/7.1	0.4 (0.2, 0.6)	15.994	<0.001	0.3 (0.1, 0.5)	7.979	0.005
*p*	0.037	<0.001	<0.001								
Carbonated drinks may increase risks of bone in child.											
Intervention Group	197/40.2	411/83.9	431/88.0	214/43.7	234/47.8	2.1 (1.8, 2.3)	217.081	<0.001	2.4 (2.1, 2.7)	201.966	<0.001
Control Group	165/45.5	198/54.5	195/53.7	33/9.0	30/8.2	0.4 (0.2, 0.5)	29.917	<0.001	0.3 (0.2, 0.5)	22.566	<0.001
*p*	0.141	<0.001	<0.001								
Fruit/vegetable drinks are not a substitute for fruits and vegetables.											
Intervention Group	276/56.3	426/86.9	447/91.2	150/30.6	171/34.9	1.6 (1.3, 1.9)	118.776	<0.001	2.1 (1.7, 2.4)	137.909	<0.001
Control Group	182/50.1	240/66.1	257/70.8	58/16.0	75/20.7	0.7 (0.5, 0.8)	55.224	<0.001	0.9 (0.7, 1.1)	71.772	<0.001
*p*	0.083	<0.001	<0.001								
Milk drinks are not a substitute for milk.											
Intervention Group	193/39.4	347/70.8	372/75.9	154/31.4	179/36.5	1.3 (1.1, 1.6)	107.327	<0.001	1.6 (1.3, 1.8)	140.814	<0.001
Control Group	126/34.7	191/52.6	207/57.0	65/17.9	81/22.3	0.7 (0.5, 1.0)	43.263	<0.001	0.9 (0.7, 1.2)	56.205	<0.001
*p*	0.174	<0.001	<0.001								
SSBs are one of the high-sugar foods.											
Intervention Group	413/84.3	467/95.3	473/96.5	54/11.0	60/12.2	1.3 (1.0, 1.7)	50.371	<0.001	1.6 (1.2, 2.1)	47.587	<0.001
Control Group	305/84.0	314/86.5	322/88.7	9/2.5	17/4.7	0.2 (−0.0, 0.4)	3.545	0.060	0.4 (0.1, 0.7)	8.855	0.003
*p*	0.925	<0.001	<0.001								
Definition of added sugar.											
Intervention Group	117/23.9	362/73.9	420/85.7	245/50.0	303/61.8	2.2 (2.0, 2.4)	302.800	<0.001	3.0 (2.7, 3.2)	389.027	<0.001
Control Group	88/24.2	204/56.2	212/58.4	116/32.0	124/7.8	1.4 (1.2, 1.6)	138.515	<0.001	1.5 (1.2, 1.7)	117.253	<0.001
*p*	0.935	<0.001	<0.001								
Adequate knowledge about SSBs ^f^											
Intervention Group	254/51.8	464/94.7	477/97.3	210/42.9	223/45.5	2.8 (2.4, 3.2)	182.719	<0.001	3.5 (3.0, 4.1)	146.612	<0.001
Control Group	212/58.4	296/81.5	302/83.2	84/23.1	90/24.8	1.1 (0.9, 1.4)	64.239	<0.001	1.3 (1.0, 1.6)	71.304	<0.001
*p*	0.061	<0.001	<0.001								

^a.^ Correctly answered. ^b.^ SSBs = sugar-sweetened beverages. ^c.^ Generalized estimating equations (GEE), controlled for child age, sex, parent’s education level, father’s BMI, mother’s BMI, physical activity time outside school (min/week), homework time (min/day), screen time (min/day) at baseline. ^d.^ β_1,_ first year compared to baseline. ^e.^ β_2_, second year compared to baseline. ^f.^ Correctly answered six or more items out of 10 above.

**Table 4 nutrients-16-00953-t004:** Family environment with parents about SSBs between the Intervention Group and Control Group before and after intervention (N ^a^/%).

Family Environment with Parents	Baseline	First Year	Second Year	Change from Baseline		Compared to Baseline ^c^					
				First Year	Second Year	β_1_ ^d^ (95%CI)	Wald_χ2_	*p*	β_2_ ^e^ (95%CI)	Wald_χ2_	*p*
My home always has SSBs ^b^											
Intervention Group	63/12.9	36/7.3	38/7.8	−27/−5.6	−25/−5.1	−0.6 (−1.0, −0.2)	8.256	0.004	−0.6 (−1.0, −0.2)	7.636	0.006
Control Group	54/14.9	49/13.5	45/12.4	−5/−1.4	−9/−2.5	−0.1 (−0.5, 0.3)	0.316	0.574	−0.2 (−0.6, 0.2)	1.051	0.305
*p*	0.421	0.004	0.026								
Parents have been warned about harms of SSBs											
Intervention Group	290/59.2	408/83.3	454/92.7	118/24.1	164/33.5	1.2 (1.0, 1.5)	99.418	<0.001	2.2 (1.8, 2.5)	136.718	<0.001
Control Group	252/69.4	296/82.2	310/85.4	44/12.8	58/16.0	0.7 (0.3, 1.0)	16.563	<0.001	0.9 (0.6, 1.3)	28.210	<0.001
*p*	0.002	0.713	0.001								
My parents restricted me from drinking SSBs											
Intervention Group	282/57.6	410/83.7	445/90.8	128/26.1	163/33.2	1.3 (1.1, 1.6)	117.524	<0.001	2.0 (1.7, 2.3)	133.415	<0.001
Control Group	205/56.5	289/79.6	312/86.0	84/23.1	107/29.5	1.1 (1.1, 1.6)	54.764	<0.001	1.6 (1.2, 1.9)	76.188	<0.001
*p*	0.780	0.150	0.029								
My parents restricted me from eating sugary snacks											
Intervention Group	260/53.1	380/77.6	429/87.6	120/24.5	169/34.5	1.1 (0.9, 1.4)	76.147	<0.001	1.8 (1.5, 2.1)	138.975	<0.001
Control Group	199/54.8	297/81.8	306/84.3	98/27.0	107/29.5	1.3 (1.0, 1.6)	62.450	<0.001	1.5 (1.1, 1.8)	71.582	<0.001
*p*	0.627	0.146	0.192								
My parents often drink SSBs											
Intervention Group	51/10.4	20/4.1	18/3.7	−31/−6.3	−33/−6.7	−1.0 (−1.5, −0.5)	16.670	<0.001	−1.1 (−1.7, −0.6)	16.175	<0.001
Control Group	34/9.4	25/6.9	28/7.7	−9/−2.5	−6/−1.7	−0.3 (−0.8, 0.2)	1.645	0.200	−0.2 (−0.7, 0.3)	0.691	0.406
*p*	0.645	0.088	0.013								
My parents often eat sugary snacks											
Intervention Group	33/6.7	40/8.2	23/4.7	7/1.5	−10/−2.0	0.2 (−0.3, 0.7)	0.709	0.400	−0.4 (−0.9, 0.2)	1.970	0.167
Control Group	27/7.4	27/7.4	35/9.6	0/0	8/2.2	−0.1 (−0.6, 0.6)	<0.001	1.000	0.3 (−0.2, 0.8)	1.139	0.286
*p*	0.687	0.797	0.006								

^a.^ Yes. ^b.^ SSBs = sugar-sweetened beverages. ^c.^ Generalized estimating equations (GEE), controlled for child age, sex, parent’s education level, father’s BMI, mother’s BMI, physical activity time outside school (min/week), homework time (min/day), screen time (min/day) at baseline. ^d.^ β_1,_ first year compared to baseline. ^e.^ β_2,_ second year compared to baseline.

**Table 5 nutrients-16-00953-t005:** The consumption of sugar-sweetened beverages (SSBs) between the Intervention Group and Control Group before and after intervention.

SSB Consumption	Baseline	First Year	Second Year	Change from Baseline		Compared to Baseline ^b^					
				First Year	Second Year	β_1_ ^c^ (95%CI)	Wald_χ2_	*p*	β_2_ ^d^ (95%CI)	Wald_χ2_	*p*
Frequency of SSB ^a^ consumption (times/week)											
Intervention Group	3.5 ± 3.4	2.4 ± 2.7	2.7 ± 3.1	−1.0 ± 4.5	−0.8 ± 4.9	−1.4 (−1.4, −0.6)	26.747	<0.001	−0.8 (−1.2, −0.4)	12.723	<0.001
Control Group	3.5 ± 3.2	3.4 ± 3.0	3.0 ± 3.2	−0.1 ± 3.9	−0.5 ± 4.0	−0.1 (−0.5, 0.3)	0.437	0.509	−0.5 (−0.9, −0.0)	4.784	0.029
T/Z	−0.144	−4.874	−1.682	−2.249	−0.765						
*p*	0.885	<0.001	0.093	0.025	0.444						
Quantity of SSB ^a^ consumption (mL/week)											
Intervention Group	871 ± 1347	638 ± 822	748 ± 1125	−233 ± 1392	−122 ± 1630	−232 (−356, −110)	13.725	<0.001	−122 (−267, 22)	2.769	0.096
Control Group	908 ± 956	1015 ± 1065	939 ± 1304	107 ± 1286	31 ± 1509	107 (−252, 239.0)	2.518	0.113	31 (−124, 186)	0.151	0.698
T/Z	−0.450	−5.835	−2.283	−2.719	−0.092						
*p*	0.653	<0.001	0.023	0.005	0.927						

^a.^ SSBs = sugar-sweetened beverages. ^b.^ Generalized estimating equations (GEE), controlled for child age, sex, parent’s education level, father’s BMI, mother’s BMI, physical activity time outside school (min/week), homework time (min/day), screen time (min/day) at baseline. ^c.^ β_1_, first year compared to baseline. ^d.^ β_2_, second year compared to baseline.

## Data Availability

The data presented in this study are available from the corresponding authors on reasonable request.

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
