# Peer review of "Sustaining Healthy Habits: The Enduring Impact of Combined School–Family Interventions on Consuming Sugar-Sweetened Beverages among Pilot Chinese Schoolchildren"

_nutrients, 2024, doi:10.3390/nu16070953_

Round 1

Reviewer 1 Report

Comments and Suggestions for Authors

Results from the study reported in this manuscript contribute to the literature about potential effectiveness of school-based nutrition interventions and potential involvement of parents. It provides some insights for the design of future interventions. Overall, the manuscript is clearly written. The Discussion addresses key results and conclusions are appropriate. There are several points throughout that need some elaboration as noted below. 

Line 54: I suggest moving the phrase “compared with infrequent SSB consumption” to either the beginning or the end of the sentence.

Line 55: I suggest adding the word “observations” after the word “These”.

In section 2.3 (Intervention methods), please indicate if parental attendance at the health sessions and receiving messages via social media channels was optional. Please elaborate in the Results section on the involvement of parents to the extent you have this information, e.g. what was the participation rate by parents at health education sessions; what percentage opted to receive messages etc.

Also, in lines 132-135, it is unclear what is meant by implementation being verified by photos and videos. What was verified? – Attendance? Presentation of the intervention? Consistency of delivery of the intervention?

Line 137: Change the word “administrated” to “administered”.

Lines 153-161: Could the 8 beverage categories be listed? Also, how did students record average amounts? Were they provided with pictures to assist with recording amounts?

Lines 288-291:  Is this study referenced in regard to the substantial and significant increases in knowledge among the control group over the 2 years? (though the increases among the intervention group were significantly greater). Were the interventions being referenced conducted in the same county as this intervention?  I think it is noteworthy that although there were increases in knowledge among the Control group, there was no change in the amounts consumed. Perhaps comment on this in relation to the differences between groups in the responses on the family environment questionnaire and any other pertinent aspects of the intervention.

Lines 324-335: Do authors have any specific insights or suggestions for future interventions based on their experience with this project? 

Comments on the Quality of English Language

The quality of the English writing is fair. The manuscript needs to be checked for proper use of verbs and verb tense as well as sentence structure. 

Author Response

Dear Reviewer:

Thank you for your and reviewer’s comments for our manuscript entitled ‘Sustaining healthy habits, the enduring impact of combined school-family interventions on the consumption of sugar-sweetened beverages among Chinese schoolchildren (No.: nutrients-2883200)’. Those comments are very helpful for revising and improving our manuscript. We revised the manuscript with tracked changes and point-to-point responses to each of comments below.

Reviewers' comments and point-to-point responses:

Reviewer #1:

Q1. Line 54: I suggest moving the phrase “compared with infrequent SSB consumption” to either the beginning or the end of the sentence.

Re: Thank you for your suggestion. The phrase has been moved to the beginning of the sentence (please see page 2, lines 51-52).

Q2. Line 55: I suggest adding the word “observations” after the word “These”.

Re: Thank you for your suggestion. The word “observations” has been added after the word “These” (please see page 2, lines 55).

Q3. In section 2.3 (Intervention methods), please indicate if parental attendance at the health sessions and receiving messages via social media channels was optional. Please elaborate in the Results section on the involvement of parents to the extent you have this information, e.g. what was the participation rate by parents at health education sessions; what percentage opted to receive messages etc.

Re: We gratefully appreciate for your valuable comment. Health lectures are arranged on the day of the parents' meeting every semester to promote the reduction of SSBs consumption to parents. Parents gather in the auditorium and listen to the health talk before returning to their respective classrooms for the parents' meeting. In China, one parent (must be the father or mother) is required to attend a parents' meeting every semester, and the attendance rate is 100%. Parents' WeChat group or QQ group is managed by the class teacher, which is used for home-school communication and assigning daily homework and tasks. Any message posted by the class teacher in the group requires parents to watch and reply. So the parental attendance at the health sessions and the information received by parents through social media channels was not optional.

Q4. Also, in lines 132-135, it is unclear what is meant by implementation being verified by photos and videos. What was verified? – Attendance? Presentation of the intervention? Consistency of delivery of the intervention?

Re: Sorry to make the reviewer confused. This is part of the quality control process. The implementation of the monthly school intervention was verified through photos and videos of the implementation of the intervention taken during the intervention, for example, take the photo of a 15-minute health education video and take the photo of put up pictures of posters in campus, etc. Schools were also asked to take photos and screen shots of the implementation of family interventions to verify the implementation of family-based interventions, such as take the videos or photos of  conduct parent health lectures and the screenshot of the class teacher delivering instant core messages to parents' WeChat or QQ groups through social media channels every month, etc. We have revised the expression in our revised manuscript (please see pages 3, lines 135-139).

Page 3, paragraph 135-139: The implementation of both family-based and school-based interventions was verified through photographic evidence and videos, ensuring the transparent documentation of activities such as health education sessions, poster displays, parent lectures, and social media interactions.

Q5. Line 137: Change the word “administrated” to “administered”.

Re: Thank you very much for pointing this out. We changed the word “administrated” to “administered” (please see pages 3, line 141).

Q6. Lines 153-161: Could the 8 beverage categories be listed? Also, how did students record average amounts? Were they provided with pictures to assist with recording amounts?

Re: Thank you for pointing this out. We have listed the 8 beverage categories (please see page 4, lines 160-164). The same questionnaire was self-administered by participating students, facilitated by a standard PowerPoint slide set in each of classrooms, in both the Intervention and Control Groups. In each school, the Study Team comprises of a local CDC staff, a class teacher and a health care teacher. They are trained by the principal investigator on how to conduct a questionnaire survey visually with a standard PowerPoint slide set in each class (please see page 4, lines 145-148). The questionnaire filled by students was collected and reviewed by the local CDC staff on the spot. The standard PowerPoint lists 8 different types of common beverage packaging and capacity pictures in the market to assist students with recording amounts. We have revised the expression from our revised manuscript (please see page 4, lines 159-171).

Page 4, paragraph 3: The weekly SSBs intake frequency questionnaire was administrated to obtain information on the children’s consumption of beverages. Eight broad beverage categories were categorized based on nutrient content and on China’s Beverage General Rule (GB10789-2008), included carbonated beverages, fruit/vegetable beverages, sweetened tea beverages, coffee beverages, lactobacillus milk beverages, sports beverages, plant-protein beverages and brewed beverages. The consumption of SSBs was measured on a 7-day frequency scale, using the question ‘How many times did you drink carbonated beverages in past week, commonly available in the market?’. The standard PowerPoint lists 8 different types of common beverage packaging and capacity pictures in the market to assist students with recording amounts. The frequency of SSBs consumption was calculated when either of eight categories of SSBs consumption was reported. The total amount of SSBs consumption was calculated by multiplying the frequency of intake by the average amount consumed each time.

Q7. Lines 288-291: Is this study referenced in regard to the substantial and significant increases in knowledge among the control group over the 2 years? (though the increases among the intervention group were significantly greater). Were the interventions being referenced conducted in the same county as this intervention? I think it is noteworthy that although there were increases in knowledge among the Control group, there was no change in the amounts consumed. Perhaps comment on this in relation to the differences between groups in the responses on the family environment questionnaire and any other pertinent aspects of the intervention.

Re: We greatly appreciate your insightful comment. In our study, the substantial and significant increases in knowledge among the control group over the 2 years were observed. Prior to the intervention, 254 (51.8%) and 212 (58.4%) students in the Intervention Group and Control Group had adequate knowledge about SSBs, respectively (X2 = 3.626, p = 0.061). One year later, 464 students (94.7%) and 296 students (81.5%) had adequate knowledge about SSBs in the Intervention Group and the Control Group, respectively (X2 = 37.127, p < 0.001). Two years later, 477 students (97.3%) in the Intervention Group and 302 students (83.2%) in the Control Group had adequate knowledge about SSBs (X2 = 52.708, p < 0.001). Compared to baseline, both groups showed increasing trends in adequate knowledge about SSBs over time (first year, Intervention Group X2 = 182.719, p < 0.001, Control Group X2 = 64.239, p < 0.001; second year, Intervention Group X2 = 146.612, p < 0.001, Control Group X2 = 71.304, p < 0.001), but the Intervention Group had a higher proportion of students with adequate SSBs knowledge than in the Control Group.

This cluster controlled trial was conducted among selected primary schools in Nanjing, China, where schools served as clusters. Two schools were randomly assigned into the Intervention Group and Control Group respectively. The Intervention Group received the packaged intervention, and the Control Group did not receive any intervention. Although the two schools in this study are in the same district, they are in administratively different geographical locations, with a point-to-point distance of 8 kilometers and different residents covered. Some measures were taken to reduce the contamination in this study. First, teachers of the two schools were informed in advance so that they could avoid exchanging and communicating about the research content. Second, liaise with the education department to ensure that control and intervention groups are not affected by other measures that may be relevant to the content of the study during the course of the study. Although the design of this study team tried to reduce potential contamination, it is still possible to be affected by contamination. We have revised this in limitations in our revised manuscript (please see page 13, lines 376-381).

Page 13, lines 376-381: Fifth, the Control Group received regular monitoring without active interventions in the first year, they may have been influenced by the Intervention Group's changes in behavior, although the study team tried to reduce contamination, there are possibilities of contamination given the short distance between two schools and very dynamic interpersonal interactions and communications.

In our study, the school-based and family-based interventions adopted educational and behavioral approaches that focus on improving children’s knowledge, attitude, and, subsequently, their behaviors towards SSBs consumption. There were no mandatory interventions not to allow the consumption of SSBs. Despite the effectiveness of our intervention in enhancing students' knowledge about SSBs, we observed a growing discordance between knowledge and practice over time. Nonetheless, the increase in knowledge does not inevitably lead to the change of practice. Higher awareness does not necessarily result in more favorable food choice practice. This phenomenon echoes the findings of Sligo et al., who noted that students may be aware of health-improving approaches but may face obstacles in translating that awareness into practice.

We have revised the expression from our revised manuscript (please see page 13, lines 346-359).

Q8. Lines 324-335: Do authors have any specific insights or suggestions for future interventions based on their experience with this project?

Re: Thank you for your suggestion. We have revised the manuscript (please see page 13, lines 398-401).

Page 13, lines 398-401: Future research should consider extending the duration or intensifying the intervention to further narrow the discordance between knowledge and practice and sustainably reduce SSBs intake.  

Q9. Comments on the Quality of English Language

The quality of the English writing is fair. The manuscript needs to be checked for proper use of verbs and verb tense as well as sentence structure.

Re: Thank you for your advice. We have revised the whole manuscript carefully and corrected grammar or syntax errors. In addition, we have asked several colleagues who are skilled authors of English language papers to check the English.

We hope these changes and responses are sufficient. Again, we appreciate the reviewer’s time and efforts in reviewing this manuscript. We feel honored to be able to publish our results in Nutrients.  

Yours sincerely,

Jinkou Zhao

Reviewer 2 Report

Comments and Suggestions for Authors

The paper “Sustaining healthy habits, the enduring impact of combined school-family interventions on the consumption of sugar-sweetened beverages among Chinese schoolchildren” contributes to the growth of literature for research and dietetics in the area of nutritional intervention, nutrition and health education, especially in high consumption of sugar among children.

However,  the following items should be revised:

Introduction

·     Line 66-68

There is evidence from studies in some countries that school-based nutrition and health education can effectively improve nutrition knowledge in children [18].  - This publication concerns one country. I suggest citing more research.

·     Line 91 – 93

“consumption of Chinese primary school students from an east region in China.”

The research concerned two schools. Therefore, I suggest changing it to “…consumption of pilot Chinese primary school students from an east region in China.”

Similar to the title

2.2. Study design

Line 106-107 “Two schools were randomly assigned to the Intervention Group and the Control Group.”  Please provide a random selection method.

Results

 “p “ - it should be Italic (p), similar to others,

Discussion

Line 274 “…..among Chinese primary schoolchildren” I suggest “two schools” or “pilot schools”

Conclusion

Line 348-348  Isn't the group too small? - two schools

I suggest: pilot schools

Was there free water in schools in subsequent years (after the intervention)?

Author Response

Dear Reviewer:

Thank you for your and reviewer’s comments for our manuscript entitled ‘Sustaining healthy habits, the enduring impact of combined school-family interventions on the consumption of sugar-sweetened beverages among Chinese schoolchildren (No.: nutrients-2883200)’. Those comments are very helpful for revising and improving our manuscript. We revised the manuscript with tracked changes and point-to-point responses to each of comments below.

Comments and Suggestions for Authors

Introduction

Q1. Line 66-68

There is evidence from studies in some countries that school-based nutrition and health education can effectively improve nutrition knowledge in children [18]. - This publication concerns one country. I suggest citing more research.

Re: Thank you for your advice. We did a literature search and have added references to relevant research (please see page 2, line 69).

Q2. Line 91 – 93

“consumption of Chinese primary school students from an east region in China.”

The research concerned two schools. Therefore, I suggest changing it to “…consumption of pilot Chinese primary school students from an east region in China. ”Similar to the title“

Re: Thank you very much for your advice. We have revised the expression in our revised manuscript (please see title and page 3, line 97).

Page 2-3, lines 95-97: Therefore, this study aimed to assess the long-term impact of the school-based and family-based interventions on reducing SSBs consumption of Chinese primary schoolchildern in pilot schools from an east region in China.

Q3. 2.2. Study design

Line 106-107 “Two schools were randomly assigned to the Intervention Group and the Control Group.”  Please provide a random selection method.

Re: Thank you for your suggestion. We have revised the manuscript (please see page 3, line 111).

Page 3, lines 110-111: Two schools were assigned either to the Intervention Group or the Control Group by computer random drawing method.

Q4. Results

“p “ - it should be Italic (p), similar to others,

Re: I’m very sorry about this. We have italicized all the “p” in our revised manuscript.

Q5. Discussion

Line 274 “…..among Chinese primary schoolchildren” I suggest “two schools” or “pilot schools”.

Re: Thank you for your suggestion. We have revised the expression in our revised manuscript (please see page 12, line 288).

Page 12, lines 286-288: To our best knowledge, this is the first study assessing the long-term impact of the combined school-based and family-based interventions on reducing SSBs consumption among Chinese primary schoolchildren in pilot schools [16–19, 27].

Q6.Conclusion

Line 348-348 Isn't the group too small? - two schools

I suggest: pilot schools

Re: Thank you for your suggestion. We have revised the expression in our revised manuscript (please see page 13, line 390).

Page 13, lines 388-392:  In summary, our study evaluating the long-term impact of combined school- and family-based interventions on reducing SSBs consumption among Chinese primary schoolchildren in pilot schools reveals positive effects on SSBs knowledge, family dynamics, and a decrease in consumption frequency during the first year, persisting into the second year following a one-year intervention and two-year follow-up.

Q7. Was there free water in schools in subsequent years (after the intervention)?

Re: We greatly appreciate your question. All the schools participating in our program were equipped with direct drinking water machines, which are cleaned and disinfected regularly. Free drinking water is available on campus to all students all the time.

We have revised the expression in our revised manuscript (please see page 13, lines 360-368).

Page 13, lines 360-368: Our results give some evidence regarding school-based and family-based interventions in the setting of an eastern culture and environment. The study also provide a sustainable environment of reduce SSBs consumption. For instance, all the schools participating in our program were equipped with direct drinking water machines, which are cleaned and disinfected regularly. Free drinking water is available on campus to all students all the time. The project schools also issued a long-standing document prohibiting the sale of SSBs on campus. It is expected that the adoption and sustaining of healthy behaviors would be facilitated through a more conducive environment and enforced by relevant policy.

We hope these changes and responses are sufficient. Again, we appreciate the reviewer’s time and efforts in reviewing this manuscript. We feel honored to be able to publish our results in Nutrients.

Yours sincerely,

Jinkou Zhao

E-mail: jinkouzhao@hotmail.com

Reviewer 3 Report

Comments and Suggestions for Authors

The study aimed to evaluate the long-term impact of combined school-based and family-based interventions on reducing sugar-sweetened beverages (SSBs) consumption among Chinese schoolchildren.  Thanks for the opportunity to review this study.

This study used a randomized controlled trial design, which is considered a robust method for evaluating the effectiveness of interventions. Author’s followed the participants over a two-year period, providing insights into the enduring effects of the interventions. By combined school-based and family-based interventions, potentially leading to a more comprehensive and effective approach to reducing SSBs consumption. The intervention group showed some positive impacts. I particularly appreciate, author’s as they included the study limitations as a sub-section. However, there are certain points for author’s to consider in including them or addressing them in their manuscript.

  1. The study noted a rebound in the quantity of SSBs consumption in the second year, which suggests that the initial effects of the interventions may not have been sustained over time. Highlight any postulating possibilities to see this rebound effect.
  2. The study focused primarily on SSBs knowledge, family dynamics, and frequency of consumption as outcome measures. Other factors, such as overall diet quality or health outcomes, were not assessed. Any specific reason, as this will be a limited Outcome Measures.

3.     While the Control Group received regular monitoring without active interventions in the first year, they may have been influenced by the Intervention Group's changes in behaviour, potentially affecting the study's outcomes why author’s have not addressed this in their discussion or in limitations section.

In conclusion, while the study demonstrated some positive effects of combined school-based and family-based interventions on reducing SSBs consumption among Chinese schoolchildren, there were also limitations in terms of sustained effects and overall quantity of consumption.

Author Response

Dear Reviewer:

Thank you for your and reviewer’s comments for our manuscript entitled ‘Sustaining healthy habits, the enduring impact of combined school-family interventions on the consumption of sugar-sweetened beverages among Chinese schoolchildren (No.: nutrients-2883200)’. Those comments are very helpful for revising and improving our manuscript. We revised the manuscript with tracked changes and point-to-point responses to each of comments below.

Reviewer #3:

Comments and Suggestions for Authors

Q1. The study noted a rebound in the quantity of SSBs consumption in the second year, which suggests that the initial effects of the interventions may not have been sustained over time. Highlight any postulating possibilities to see this rebound effect.

Re: Thank you for your suggestion. We have revised the hypothesis in our revised manuscript (please see page 3, lines 92-95).

Page 3, lines 92-97: We hypothesized that after one year intervention, the knowledge and SSBs consumption of the Intervention Group would decrease, but the persistence of the intervention effect is unknown, and there may be a rebound after 1 year of intervention. Therefore, the aim of this study was to assess the long-term impact of the school-based and family-based interventions on reducing SSBs consumption of Chinese primary schoolchildren in pilot schools from an east region in China.

Q2. The study focused primarily on SSBs knowledge, family dynamics, and frequency of consumption as outcome measures. Other factors, such as overall diet quality or health outcomes, were not assessed. Any specific reason, as this will be a limited Outcome Measures.

Re: We greatly appreciate your insightful comment. We have revised limitations in our revised manuscript (please see page 13, lines 374-376).

Page 13, lines 374-376: Forth, we did not collect student’s total food intake and body measurements other than height and weight. We could not therefore assess the diet quality or health outcomes.

Q3. While the Control Group received regular monitoring without active interventions in the first year, they may have been influenced by the Intervention Group's changes in behavior, potentially affecting the study's outcomes why author’s have not addressed this in their discussion or in limitations section.

Re: We greatly appreciate your insightful comment. This cluster controlled trial was conducted among selected primary schools in Nanjing, China, where schools served as clusters. Two schools were randomly assigned into the Intervention Group and Control Group respectively. In our study, school-based and family-based interventions were packaged together. The Intervention Group received the packaged intervention, and the Control Group did not receive any intervention the two schools in this study are in the same district, they are in different geographical locations, with a linear distance of 8 kilometers and different residents covered. Some measures were taken to reduce the contamination in this study. First, teachers of the two schools were informed in advance so that they could avoid exchanging and communicating about the research content. Second, liaise with the education department to ensure that control and intervention groups are not affected by other measures that may be relevant to the content of the study during the course of the study.

Although the design of this study team tried to reduce the possibility of contamination, it is still possible to be affected by contamination. We have revised limitations in our revised manuscript (please see page 13, lines 376-381).

Page 13, lines 376-381: Fifth, the Control Group received regular monitoring without active interventions in the first year, they may have been influenced by the Intervention Group's changes in behavior, although the design of this study team tried to reduce contamination, there are possibilities of contamination given the short distance between two schools and very dynamic interpersonal interactions and communications.

We hope these changes and responses are sufficient. Again, we appreciate the reviewer’s time and efforts in reviewing this manuscript. We feel honored to be able to publish our results in Nutrients.

Yours sincerely,

Jinkou Zhao

E-mail: jinkouzhao@hotmail.com

Round 2

Reviewer 1 Report

Comments and Suggestions for Authors

Thank you for your responses and your thorough attention to the comments. All questions have been sufficiently addressed. There are two minor things:

Line 16: 'radom" should be "random".

Line 91: Did you mean to say that knowledge would INCREASE and consumption of SSBs would decrease? As it reads now, it indicates that you hypothesized that both knowledge and consumption would decrease.

Author Response

Dear Reviewer:

Thank you for your comments for our manuscript entitled ‘Sustaining healthy habits, the enduring impact of combined school-family interventions on the consumption of sugar-sweetened beverages among Chinese schoolchildren (No.: nutrients-2883200)’. Those comments are very helpful for revising and improving our manuscript. We revised the manuscript with tracked changes and point-to-point responses to each of comments below.

Reviewers' comments and point-to-point responses:

Reviewer #1:

Q1. Line 16: 'radom" should be "random".

Re: Thank you for your suggestion. The word “radom” was changed to “random” (please see page 1, line 16).

Page 1, lines 15-17: Two primary schools were assigned at random to either the Intervention Group or the Control Group, in Nanjing, eastern China.

Q2. Line 91: Did you mean to say that knowledge would INCREASE and consumption of SSBs would decrease? As it reads now, it indicates that you hypothesized that both knowledge and consumption would decrease.

Re: Sorry to make the reviewer confused and we greatly appreciate your insightful comment. We have revised the hypothesis in our revised manuscript (please see page 2, lines 91-94).

Page 2, lines 91-94: We hypothesized that after one year intervention, in the Intervention Group, the knowledge about SSBs would be sustain at a level higher than baseline values and SSBs consumption would remain lower than baseline values, while acknowledging there may be a rebound after 1 year of intervention.

We hope these changes and responses are sufficient. Again, we appreciate your time and efforts in reviewing this manuscript. We feel honored to be able to publish our results in Nutrients.  

Yours sincerely,

Jinkou Zhao
